# The Use of Human Amniotic Membrane (hAM) as a Treatment Strategy of Medication-Related Osteonecrosis of the Jaw (MRONJ): A Systematic Review and Meta-Analysis of the Literature

**DOI:** 10.3390/medicina59050968

**Published:** 2023-05-17

**Authors:** Roberto Sacco, Oladapo Akintola, Nicola Sacco, Alessandro Acocella, Monica Diuana Calasans-Maia, Massimo Maranzano, Sergio Olate

**Affiliations:** 1Oral Surgery Department, School of Medical Sciences, Division of Dentistry, The University of Manchester, Manchester M13 9PL, UK; 2FACOP—Faculdade do Centro Oeste Paulista, Dental School, Oral Surgery Department, Bauru 17499-010, Brazil; 3Oral Surgery Department, King’s College Hospital NHS Trust, London SE5 9RW, UK; 4Anaesthetic & Critical Care, University of Campania Luigi Vanvitelli, 80138 Caserta, Italy; 5Private Practice, 50129 Florence, Italy; 6Oral Surgery Department, Dental School, Fluminense Federal University, Rio de Janeiro 24020-140, Brazil; 7Oral and Maxillofacial Surgery Department, Manchester University NHS Foundation Trust (MFT), Manchester M13 9WL, UK; 8Division of Oral and Maxillofacial Surgery, Universidad de La Frontera, Temuco 4780000, Chile

**Keywords:** medication-related osteonecrosis of the jaw, skeletal-related events, human amniotic membrane, cancer, antiresorptive drugs, antiangiogenic drugs

## Abstract

*Background and objectives*: Although it is very uncommon, medication-induced osteonecrosis of the jaw (also known as MRONJ) can have serious consequences. Traditionally, this adverse event has been recognised in patients who were treated with bisphosphonate (BP) drugs. Nevertheless, in recent years, it has been established that individuals having treatment with various types of medications, such as a receptor activator of nuclear factor kappa-Β ligand inhibitor (denosumab) and antiangiogenic agents, have had the same issue. The purpose of this research is to determine if the application of human amniotic membrane (hAM) may be used as a therapy for MRONJ. *Material and Methods*: A multi-source database (MEDLINE, EMBASE, AMED, and CENTRAL) systematic search was performed. The major objective of this study is to obtain an understanding of the efficacy of hAM when it is employed as a treatment modality for MRONJ. The protocol of this review was registered in the INPLASY register under the number NPLASY202330010. *Results*: The authors were able to include a total of five studies for the quality analysis, whereas for the quantity evaluation, only four studies were eligible. A total of 91 patients were considered for the investigation. After treatment with human amniotic membrane (hAM), a recurrence of osteonecrosis was observed in n = 6 cases (8.8%). The combined efficacy of surgical therapy and the use of hAM resulted in an overall success rate of 91.2%. Intraoperative complications were only documented in one article, and they were mostly caused by the positioning of the hAM, which led to wound breakdown at the surgical site. *Conclusions:* Based on the small amount of data and low-quality research included in this study, using human amniotic membranes to treat MRONJ might represent a feasible option. Nevertheless, further studies with a wider patient population are required to understand the long-term impacts.

## 1. Introduction

Since 2003, medication-related osteonecrosis of the jaw (MRONJ) has been a major concern as a severe side effect, mainly caused by bisphosphonate (BP) therapy for both oncology and non-oncology type of patients [1]. The American Association of Oral and Maxillofacial Surgeons (AAOMS), in the following years, has published a series of positional papers in the attempt to alert and guide surgeons and clinicians about this condition [2,3,4,5].

Generally, individuals affected by MRONJ can have a sequela of considerable symptoms from mild to severe and from discomfort to a complete and impaired overall quality of life and an increased morbidity [6,7]. Unfortunately, in the past few decades, the consumption of antiangiogenic and antiresorptive agents such as bisphosphonates (BPs) and receptor activator of NF-kB ligand (RANKL) inhibitors, which are directly responsible for MRONJ, have increased due to a growing number of patients suffering from osteoporosis and malignancy across the globe [8].

Recent research publications on the subject have shown that the frequency of MRONJ in patients suffering from osteoporosis varies from 0.001 to 0.1% [1,5], while the frequency of MRONJ in patients suffering from skeletal-related cancer ranges from 0.7% to 6.7% [5,6,9]. Yet, due to the restricted research and poor study designs used in these earlier investigations, it is possible that they produced erroneous findings; hence, incidence of the disease could be much higher.

At the present time, the management of MRONJ has remained a controversial topic within the oral and maxillofacial surgery community. To date, therapeutic management has focused largely on symptomatic treatment, divided into the categories of surgical therapy and nonsurgical therapy. According to the AAOMS, MRONJ was categorised into four different stages (stages 0 to III), giving the severity of the disease [4,5]. Indeed, all the AAOMS’ positional papers that have suggested treatment strategies, including medical and surgical management, have been discussed. However, the literature has documented different kinds of treatments, from the less invasive, such as antimicrobial mouth rinses, systemic antibiotics, hyperbaric oxygen therapy, pentoxifylline, and teriparatide, to the surgical one with different degrees of invasiveness [10,11,12,13].

In more recent years, we noticed a shift in the literature for the correct management of MRONJ, with the vision of improving the prognosis and reducing the risk of recurrence. Especially, the attention has been on the use of adjuvant treatments such as the autologous platelet concentrate (APC), recombinant human bone morphogenetic protein 2 (rhBMP-2), and human amniotic membrane (hAM) [14,15,16,17].

hAM contains amniotic epithelial cells and amniotic mesenchymal stromal cells (and variable quantities of growth factors [18,19]). The beneficial effects of hAM have been widely documented. Research has described it as a bio-compatible scaffold with suitable mechanical properties (permeability, stability, elasticity, flexibility, resorbability, and transparency) [20,21]. Additionally, this allograft material possesses antifibrotic, antimicrobial, anti-inflammatory, and analgesic properties [20,22,23]. Indeed, it modulates angiogenesis, having both pro-antiangiogenic properties, and induces epithelialisation and wound healing [20,21]. Furthermore, hAM contains a large quantity of growth factors (EGF, FGF, and TGF) as well as metalloprotease inhibitors, all of which are important for tissue repairing. Its low immunogenicity as a result of having fewer human leukocyte antigens (HLAs) or b2 microglobulin peptides complements all of the aforementioned characteristics [24]. Hence, hAM appears to be a viable material to prevent and/or aidMRONJ-exposed patients.

The aim of this systematic and meta-analysis review is to investigate the available scientific evidence on the use of hAM as a treatment strategy of MRONJ.

## 2. Materials and Methods

This review was carried out in accordance with the Preferred Reporting Items for Systematic Reviews and Meta-Analyses (PRISMA) criteria and was recorded in INPLASY under the registration number INPLASY202330010 [25].

PubMed, MEDLINE, EMBASE, and CINAHL were the four databases scrutinised for this review. To ensure the quality of the searches, a three-stage focus screening strategy was applied. Two authors (RS and NS) independently screened the titles and abstracts to exclude extraneous content (i.e., reviews, animal studies, non-clinical studies, and non-randomised control studies). Disagreements were settled via discussions with a different third and fourth author (MDCM and SO) before an agreement was reached.

A form for data filtering and abstraction was employed to:Confirm study eligibility based on inclusion and exclusion criteria.Conduct methodical quality assurance.Gather information on study features and results.

For those studies suitable for inclusion in the review but lacking adequate information, the corresponding authors were contacted. The PICOS framework was used to break down and develop clinical questions [26].

Focused question and PICOS strategy.

Is the human amniotic membrane a viable and successful treatment option for patients affected by MRONJ?

Population (P): MRONJ patients of any age (no age restrictions).Interventions (I): any MRONJ therapy (including surgical and non-surgical) using human amniotic membrane.Comparison (C): not applicable.Outcome (O): MRONJ healing and recurrence.Study design (S): Randomised controlled trials, case-controlled trials, cohort studies (prospective and or retrospective), case series, and case report.

Figure 1 summarises the search and the article selection strategy.

### 2.1. Criteria for Inclusion in this Review

Study type: The research strategy comprised available or unpublished randomised controlled trials, case-controlled trials, case series, retrospective research, and case reports. From January 2003 to March 2023, publications were collected. Animal studies, reviews, and research, including people who had already had radiation treatment to the head and neck areas, were all removed. The search was not restricted by language.

Types of Participants: The systematic review aimed to include any type of study involving individuals (oncology and non-oncology patients) with MRONJ who were treated with human amniotic membranes, with no age restrictions. There was no limit on the number of patients who could participate in the research.

Objectives: The main purpose of this research was to evaluate the ability of the human amniotic membrane in the treatment of jaw osteonecrosis.

### 2.2. Outcomes Measured

Primary outcome: Determine the success rate (full healing) of treating MRONJ patients using human amniotic membrane.

Secondary outcome: Consider the following factors:The most frequent dental therapy associated with MRONJ (dental extraction, spontaneous, denture, etc.);The common MRONJ anatomical location (maxilla, mandible, anterior, or posterior of the jaws);MRONJ recurrence rate;The most often occurring location.

### 2.3. Data Extracted

The Cochrane Public Health Group Data Extraction and Evaluation Template was used to retrieve relevant data. All of the publications that were chosen were thoroughly examined to determine the author(s), year of publication, research design, demographics, and treatment features. The number of patients, patient details (including age and gender), type of drug-related necrosis, type of complications, and hAM success rate were all retrieved from the research. The data collected was recorded into a predetermined and structured Microsoft Excel form. In the event of missing data, authors were informed and given six weeks to respond. If the data was still unavailable, it was marked as ‘not reported (NR)’ in the outcomes and tables.

### 2.4. Statistical Analysis

The study’s main goal was to evaluate the recurrence of MRONJ in individuals treated with surgical ablation and hAM. For quantitative variables, the results were reported as means and standard deviation (SD), and for categorical data, as numbers and proportions. The Q-statistic and the accompanying I2 coefficient were used to measure the studies’ heterogeneity. The results were deemed significant at the 5% level (*p* < 0.05). If there was heterogeneity (Q-test *p*-value < 0.10), effect estimates were pooled using a random-effects model. All the studies included in this review were evaluated using a specific meta-analysis software programme (JAMOVI; The Jamovi Project, 2021). A forest and a funnel plot were used to illustrate the meta-analysis.

### 2.5. Review of Risk of Bias and Assessment of Quality of Evidence

Two review authors (RS and SO) assessed the risk of bias of the relevant, included research using the criteria provided by the Cochrane Handbook for Systematic Reviews of Interventions as relevant for randomised clinical trials (RCTs) and non-randomised clinical trials (non-RCTs). Any disputes in the risk of bias evaluations were submitted to another review team author (NS) and addressed via conversation [27,28]. The GRADE profiler (GRADEpro) programme was used to evaluate the quality of evidence and summarise the results, as recommended by the Cochrane Collaboration and the GRADE Working Group [29].

## 3. Results

This review included a total of five publications. All of the available results describe subjects treated between 2018 and 2022. Case series (n = 3; 60%), prospective studies (n = 1; 20%), and randomised clinical trials (n = 1; 20%) were included in this research review. Table 1 summarises the features of the research examined in this review [17,30,31,32,33].

### 3.1. Patients and MRONJ Data Analysis

A total of 91 participants met the inclusion criteria for the five articles selected. In four of the five investigations, the mean age of the participants was 68.15 ± SD 12.21 years (range 36–89 years). The patient population included 69 female patients (75.8%) and 22 male patients (24.2%). All patients were given antiresorptive and/or antiangiogenic therapies or a combination of both. The majority of MRONJ patients were oncologic (n = 60; 65.9%), followed by non-oncologic (n = 29; 31.9%). Only one article (n = 2; 2.2%) failed to identify the kind of medical condition affecting the patient. Bisphosphonate was the most frequently identified bone target drug with MRONJ (n = 58; 63.7%). Nevertheless, no information was supplied in 29.7% (n = 27) of the events. The length of bone target treatment prior to MRONJ formation was only clearly documented in 1 out of 5 papers, with a mean of 39 ± SD 34.69 months (ranging from 11 to 101 months). Just two of the five publications indicated the MRONJ triggering event. The mandible was the most prevalent MRONJ site (n = 55; 60.4%), followed by the maxilla (n = 27; 29.7%). Just nine (9.9%) of the patients exhibited MRONJ in both of the jaws. Moreover, data on the MRONJ stage was obtained using the AAOMS definition. The most prevalent MRONJ stage among the 91 patients included in the research was stage 2 (n = 50; 54.9%), followed by stage 1 (n = 30; 33%) and stage 3 (n = 11; 12.1%). All subjects (n = 68 of the 91; 74.7%) were treated with human amniotic membrane and were followed up on for 3 to 42 months. MRONJ recurrence was detected in only n = 6 patients, yielding an incidence of 8.8% and a total success rate of 91.2% (n = 57). N = 5 (7.3%) of the patients saw an improvement in the necrotic region. All recurrences were discovered within 1 to 6 months of the follow-up. Table 2 summarises the information gathered from the studies.

### 3.2. Meta-Analysis Evaluation

Based on the inclusion criteria for this review, only four out of five studies were suitable for meta-analysis; all publications included had a minimum of 3 to 42 months of follow-up. N = 6 MRONJ recurrent cases were detected among the patients treated with hAM. The meta-analysis included the incidence of MRONJ, which was pooled using a random-effects model method. The results of the study revealed a significant overall impact [*p* = 0.010; Z = 2.58] as well as heterogeneity [*p* = 0.949; df: 3.000; I2: 0%]. The raw proportion (RP) model was used to calculate the effect size measures (0.09) with a 95% confidence interval (CI: 0.02–0.15). Figure 2 and Figure 3 and Table 3 and Table 4 summarise the meta-analysis evaluation. Only one publication out of five reported the frequency of post-operative problems (excluding MRONJ recurrence) in patients receiving human amniotic membrane, reporting solely issues related to the placement and adaptation of the hAM to the surgical site.

Only one publication out of five reported the frequency of post-operative problems (excluding MRONJ recurrence) in patients receiving human amniotic membrane, reporting solely issues related to the placement and adaptation of the hAM to the surgical site.

### 3.3. Risk of Bias Assessment

The risk of bias of the included studies are illustrated in Figure 4 and Figure 5. Due to the nature of the studies, the risk of assessment bias was performed using RoB 2 and ROBINS-I.

Randomisation—Ragazzo et al. stated in their paper that the participants were randomly allocated [32]. The mechanism of sequence creation, however, was not mentioned. As a result, the degree of bias was deemed uncertain. All of the other studies included in this review were not randomised; hence, all of the publications were assessed to be at high risk of bias.

Intervention and blinding bias—Due to the lack of a blinding mechanism, the risk of bias due to deviations from the planned treatments was deemed high in all the studies.

Missing data bias—The risk of bias related to incomplete outcome data was deemed high in all of the studies. This was due to a lack of sample size and effect size calculations. Furthermore, some studies showed a significant percentage of dropouts.

Outcome bias—The risk of bias in outcome measurement was evaluated in all research. Moreover, owing to the large number of dropouts, Ragazzo et al. and Val et al. were deemed to have a high risk of bias [32,33].

Reporting bias—All the studies included in the research reported to have a high risk of bias in selection.

Overall bias ad quality—The overall quality of the studies included in this review were considered poor.

### 3.4. Review Quality Assessment

To evaluate the quality of the evidence, the 5 GRADE criteria (study limitations, consistency of effect, imprecision, indirectness, and publication bias) were utilised. Using the GRADEpro software programme (GRADEpro GDT, 2015), a “summary of findings” table was created to capture the effect and quality of evidence for each outcome (Table 5). In terms of the overall quality of the evidence offered by the chosen studies, all of them were graded as poor quality, owing to a high risk of bias since each research study had at least one area rated as high risk. As a consequence, the estimations of the influence of the findings are imprecise, and the reported results should be evaluated with caution.

## 4. Discussion

MRONJ has been studied and classified by several organisations throughout the globe. Opinions and disease management strategies based on stage-specific approaches have been offered; however, no consensus agreement has been found [5,34,35]. Nonetheless, according to several standards, stages 1 and 2 are considered to be conservatively managed, whereas surgery is encouraged for severe cases. In a recently published systematic review on the use of radical surgical intervention and free tissue graft transplant in MRONJ cases, it was reported that many patients with advanced stages of the illness may be unable to be treated due to the advancement of the primary disease, leaving patients with a reduced quality of life [12].

Several research studies have shown that surgical intervention in MRONJ has a much higher success rate than a pure pharmaceutical therapy strategy [36,37]. According to a previous study, the MRONJ therapeutic goal should be pain relief, controlling diffused tissue infection, and minimising the progression of osteonecrosis [38]. Due to the special characteristics of the oral cavity as well as antiresorptive medication treatment, this constitutes a significant problem. It is generally recognised that microbial contamination is increased in MRONJ patients, particularly those on antibiotic treatment [39]. Moreover, antiresorptive medicines may cause soft tissue harm by reducing epithelial cell proliferation and increasing cellular death [4,5]. Therefore, it becomes extremally challenging to close any bone defect or exposure. In this regard, hAM has anti-inflammatory characteristics, which are conserved even during a cryopreservation process at −80 °C [20,23,40,41]. Furthermore, hAM has antimicrobial properties, which are attributable to the presence of bactricidin, beta-lysin, lysozyme, transferrin, and 7-S immunoglobulins in the amniotic fluid [42]. Hence, hAM could be a viable option of treatment due to its properties.

hAM is the innermost layer of the placenta that surrounds the developing foetus during pregnancy. It is a thin, transparent membrane that contains a variety of biological components that have been shown to possess several unique features and therapeutic properties. hAM has a high degree of biocompatibility, meaning that it is well-tolerated by the human body and can be used as a scaffold for tissue engineering and regenerative medicine applications. This makes it useful for treating conditions such as burns, wounds, and certain autoimmune diseases. However, very few studies have tested this allograft as an adjuvant therapeutic agent in the case of osteonecrosis of the jaws [20,23].

Our findings have shown that hAM has a very high degree of success rate of 91.2%. Additionally, data have also shown an additional 50% of MRONJ improvement. However, this data was only reported in one article. Complications were reported only intra-operatively in one article. The common complication was associated in positioning the hAM in the surgical site [31].

The same research article reported that when non-primary closure of the surgical site was achieved with hAM, the allograft was completely resorbed within the two weeks.

Three out of five articles investigated the pain symptoms associated before and after the procedure [31,32,33].

Odet et al. measured pain after surgery at every follow-up time (on days 7 and 14 and at months 1, 2, 3, and 6) using the Visual Analogue Scale (VAS). However, no specification on the type of evoking stimulus measurement was given. Meanwhile, for Val et al. and Ragazzo et al., pain was measured using the VAS from 0 to 10 at three different times: at rest, during feeding, and phonation [32,33]. These measurements were presented at three different times of the investigation: before the intervention (T0), 7 (T1), and 30 (T2) days after the operation. Both articles indicated a statistically significant (*p* = 0.028) pain improvement across studies [32,33]. Val et al. reported 92.5% of pain improvement within the first week of the surgical procedure [33]. Furthermore, three out of five articles described the nature of the special investigation during the follow-up time. Orthopantomography paired with CBCT were used by Çanakçi et al. and Odet et al. [30,31]. Ragazzo et al. described using only orthopantomography during the follow-up times [32].

According to the results shown by this review, additional and robust studies are necessary to confirm the validity of the hAM as an adjuvant treatment for MRONJ.

In general, the authors advocate that the following rules should be applied in the research protocols:Large cohort studies with a sample size calculation should be carried out and described in sufficient detail to verify a representative success rate of using this treatment modality.Common and reliable measurable studies’ endpoints including the negative outcomes should be described in a sufficiently detailed manner.A patient’s risk stratification is essential in understanding the rate of success and complications in different risk categories.A reliable follow-up examination other than clinical (e.g., CT, CBCT, or MRI) should be used to identify local MRONJ recurrence.

## 5. Conclusions

Based on a multiple-reviewer quality assessment criteria, this systematic review was only capable of identifying publications that met Levels 2b and 4 of the Oxford Evidence-Based Medicine Hierarchies. Despite part of the study being ranked as poor quality, the hAM seems to be a useful surgical adjuvant therapy for MRONJ. In fact, fifty-seven patients out of sixty-eight had a full recovery. The majority of patients had considerable alleviation from pain and infectious signs shortly after surgery. Nevertheless, cautious optimism about the positive outcomes is required, and more quality research, such as control multicenter studies or randomised clinical trials, is required to corroborate and confirm the review’s conclusions.

Much further study is required to completely understand the therapeutic mechanism and appropriate dose of hAM in the setting of osteonecrosis, but the available information shows that it might be a useful therapy alternative in the management of this disorder.

## Figures and Tables

**Figure 1 medicina-59-00968-f001:**
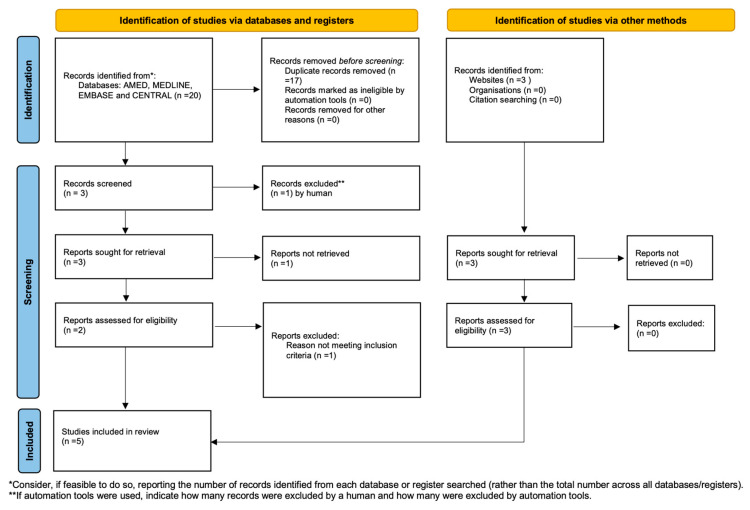
Preferred Reporting Items for Systematic Reviews and Meta-Analyses (PRISMA) flowchart.

**Figure 2 medicina-59-00968-f002:**
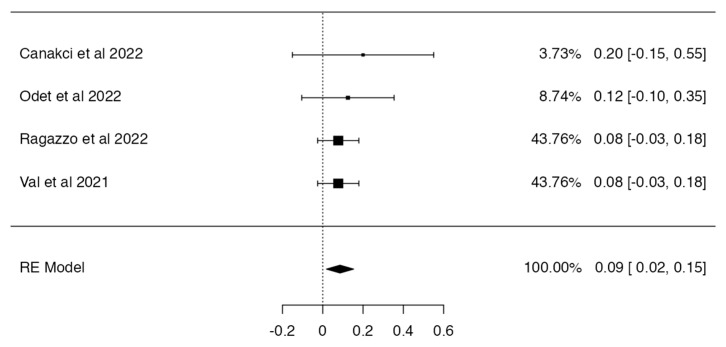
Forest plot [30,31,32,33].

**Figure 3 medicina-59-00968-f003:**
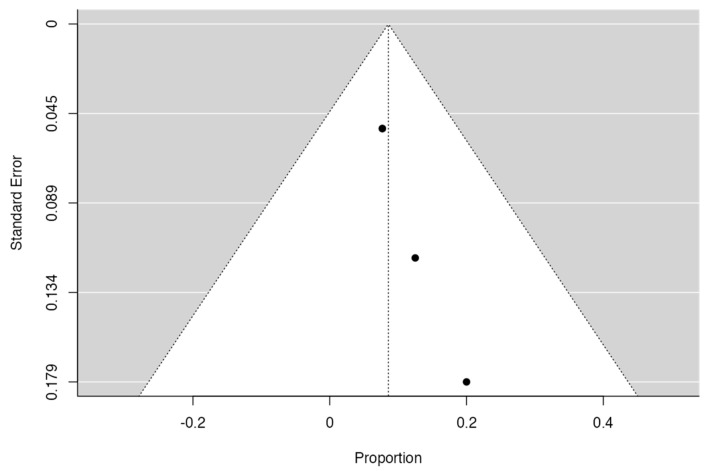
Funnel plot.

**Figure 4 medicina-59-00968-f004:**
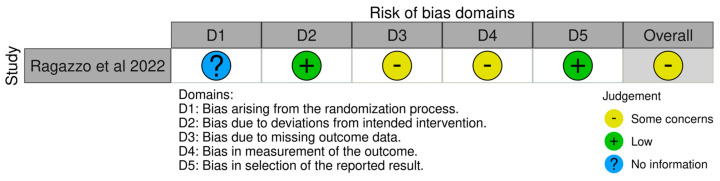
RoB 2 assessment [32].

**Figure 5 medicina-59-00968-f005:**
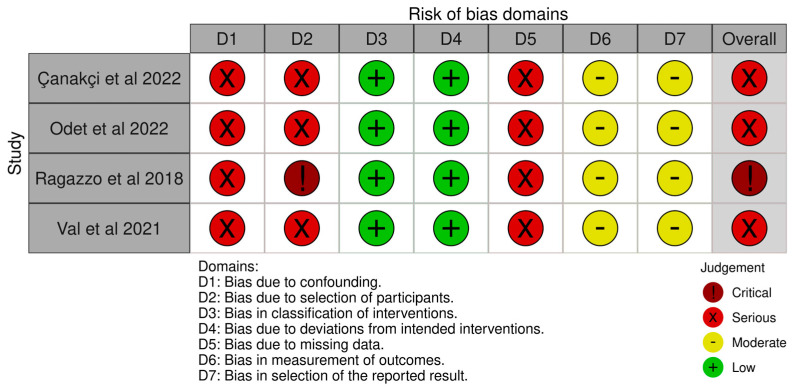
ROBINS-I assessment [17,30,31,33].

**Table 1 medicina-59-00968-t001:** Summary of the included studies’ characteristics [17,30,31,32,33]. Study level according the Oxford Centre for Evidence-Based Medicine: Levels of Evidence (March 2009).

Authors	Study Design	Study Population	Source of Recruited Participants	Inclusion Criteria	Study Level
Çanakçi et al. 2022	CS	5	University	History or currently taking antiresorptive or antiangiogenic with MRONJ	4
Odet et al. 2022	CS	8	University /Hospital	History or currently taking antiresorptive or antiangiogenic with MRONJ stage 2	4
Ragazzo et al. 2018	CS	2	Hospital	History or currently taking antiresorptive or antiangiogenic with MRONJ	4
Ragazzo et al. 2022	RCT	26/23	Hospital	History or currently taking antiresorptive or antiangiogenic with MRONJ any stages	2b
Val et al. 2021	CS	26	Hospital	History or currently taking antiresorptive or antiangiogenic with MRONJ any stages	2b

**Table 2 medicina-59-00968-t002:** Summary of the studies found [17,30,31,32,33]. Bisphosphonate (BP); Denosumab (DMAB); Osteoporosis (OP); Multiple Myeloma (MM); Breast Cancer (BC); Prostate Cancer (PC); Lung Cancer (LC); Thyroid Cancer (TC); Rheumatoid Arthritis (RA); Leukemia (Leuk); Renal Cancer (RC); Not Reported (NR).

Authors	Age (Mean-Years)	Males/Females (M/F Ratio)	Type of Bone Targeting Drug Therapy	Drug Time Length (Mean-Months)	Medical Condition Associated to the Bone Targeting Drug Therapy	MRONJ Site	Complications	MRONJ Recurrence
Çanakçi et al. 2022	Mean 62 (range 42–82)	3 (M)/2 (F)	5 × BP	Mean 39 (range 11–101)	3 × PC 2 × BC	3 × Mandible 2 × Maxilla	NR	n = 1
Odet et al. 2022	Mean 67.5 (range 49–88)	2 (M)/6 (F)	3 × BP + DMAB 2 × BP 2 × DMAB 1 × Antiangiogenic	NR	4 × BC 2 × MM 1 PC 1 × RC	7 × Mandible 1 × Both	4 intra-op	n = 1
Ragazzo et al. 2018	Mean 85	1 (M)/1 (F)	2 × BP	NR	NR	2 × Mandible	NR	n = 0
Ragazzo et al. 2022	Mean 68.3 (range 36–89)	10 (M)/39 (F)	49 × BP	Unclear	15 × OP 13 × MM 11 × BC 3 × PC 2 × LC 2 × TC 1 × RA 1 × LC + Leuk 1 × Algodystrophy	26 × Mandible 18 × Maxilla 5 × Both	NR	n = 7 (n = 2 × hAM group)
Val et al. 2021	68.4 (range 36–89)	6 (M)/21 (F)	Unclear	NR	15 × Cancer 10 × OP 1 × RA 1 × Algodystrophy	17 × Mandible 7 × Maxilla 3 × Both	NR	n = 2

**Table 3 medicina-59-00968-t003:** Meta-analysis with random-effects model (k = 4).

	Estimate	se	Z	*p*	CI Lower Bound	CI Upper Bound
Intercept	0.0857	0.0346	2.48	0.013	0.018	0.153

Note. Tau² Estimator: Restricted Maximum-Likelihood.

**Table 4 medicina-59-00968-t004:** Heterogeneity statistics.

Tau	Tau²	I²	H²	R²	df	Q	*p*
0.000	0 (SE= 0.0037)	0%	1.000	.	3.000	0.578	0.902

**Table 5 medicina-59-00968-t005:** Summary of findings [17,30,31,32,33]. GRADE Working Group grades of evidence.

Study	Quality Assessment					Grade of Evidence
Authors	Risk of Bias	Inconsistency	Indirectness	Imprecision	Other Considerations	Quality
Çanakçi et al. 2022	Serious ^a^	Serious ^b^	Serious ^c^	Serious ^d^	None	⊕⊕⊝⊝ Low
Odet et al. 2022	Not serious	Serious ^b^	Serious ^c^	Not serious	None	⊕⊕⊝⊝ Low
Ragazzo et al. 2018	Not serious	Serious ^b^	Not serious	Not serious	None	⊕⊕⊝⊝ Low
Ragazzo et al. 2022	Not serious	Not serious	Serious ^c^	Serious ^d^	None	⊕⊕⊕⊝ Moderate
Val et al. 2021	Not serious	Not serious	Not serious	Not serious	None	⊕⊕⊝⊝ Low

High: We are very confident that the true effect lies close to that of the estimate of the effect. Moderate: We are moderately confident in the effect estimate: The true effect is likely to be close to the estimate of the effect, but there is a possibility that it is substantially different. Low: Our confidence in the effect estimate is limited. The true effect may be substantially different from the estimate of the effect. Very low: We have very little confidence in the effect estimate. The true effect is likely to be substantially different from the estimate of effect. ^a^ Quality of evidence downgraded one level on the basis of unclear risk of selection bias. ^b^ Quality of evidence downgraded one level on the basis of heterogeneity. ^c^ Quality of evidence downgraded one level on the basis of indirectness. ^d^ Quality of evidence downgraded one level on the basis of imprecision.

## Data Availability

Data are included in the article.

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
