# Peer review of "The Use of Human Amniotic Membrane (hAM) as a Treatment Strategy of Medication-Related Osteonecrosis of the Jaw (MRONJ): A Systematic Review and Meta-Analysis of the Literature"

_medicina, 2023, doi:10.3390/medicina59050968_

Round 1

Reviewer 1 Report

Dear Authors,

First of all, I would like to thank you for submitting a manuscript about this specific topic. This is a well-written and organized systematic review. In my opinion, this manuscript is highly valuable, especially for clinicians who are performing MRONJ-related surgical operations. Furthermore, such studies might be beneficial for future position papers about this clinical situation.

Author Response

Reviewer No 1

Dear Authors,

First of all, I would like to thank you for submitting a manuscript about this specific topic. This is a well-written and organized systematic review. In my opinion, this manuscript is highly valuable, especially for clinicians who are performing MRONJ-related surgical operations. Furthermore, such studies might be beneficial for future position papers about this clinical situation.

Answer: Dear reviewer thanks for your words of encouragement. We believe that the aim of every researchers is to help the community through shearing new knowledge and findings.

Reviewer 2 Report

The study is well conducted and could improve the quality of the clinical research in the use of amniotic membrane for MRONJ.

I suggest the following corrections:

- line 64: please correct the citations;

- line 84: please correct the subject of the sentence;

- lines 103-117: I think that these sentences must be removed.

Author Response

Reviewer No2

The study is well conducted and could improve the quality of the clinical research in the use of amniotic membrane for MRONJ.

I suggest the following corrections:

- line 64: please correct the citations;

Answer: Thanks for your remark. We have reviewed the citations and correct them as per suggestion.

- line 84: please correct the subject of the sentence;

Answer: Thanks for your remark. We have reviewed the sentences and improve the readability

- lines 103-117: I think that these sentences must be removed.

Answer: Thanks for your remark. We have removed the sentences as suggested

Reviewer 3 Report

The review is very well done and clear.

Even seems very strict in the papers' evaluation, It encourage creating awareness of the possibility to use Human Amniotic Membrane in MRONJ, this is very important.

I cannot see the Fig. 1-4-5, they are empty squares in the downloaded pdf so I am not able to review them.

In the discussion, paragraph starting from line 349 seems to be an introduction of the tissue and should be at the beginning of the topic.

Kind Regards

Author Response

The review is very well done and clear.

Even seems very strict in the papers' evaluation, It encourage creating awareness of the possibility to use Human Amniotic Membrane in MRONJ, this is very important.

Answer: Thanks for your feedback and words of encouragement.

I cannot see the Fig. 1-4-5, they are empty squares in the downloaded pdf so I am not able to review them.

Answer: Thanks for your remark. I have received a note from the Managing Editor who has informed us that the problem with the figures was resolved. Hence, I believe that eventually these figures were reviewed.

In the discussion, paragraph starting from line 349 seems to be an introduction of the tissue and should be at the beginning of the topic.

Answer: Thanks for your comment. After a discussion among the authors, we believed that the paragraph starting at line 349 should be left as it is. As you could appreciate, the characteristics and features of the material have already been described in the introduction (please refer to lines 82 to 94). The paragraph has been inserted in a specific place in order to emphasise the lack of knowledge of the hAM therapeutic application in MRONJ cases, despite its intrinsic biological features.

Kind Regards